# Explainable Generative Multi-Modality Alignment for Transferable Recommendation

Submission Id: 2352*

## Abstract

With the development of multi-modality data modeling techniques, recent recommender systems use not only textual data and user-item interactions but also multi-modality data such as images to improve their performances. Existing methods typically adopt cross-modal pairwise alignment strategies to alleviate the gap between modalities. Nevertheless, this alignment paradigm has limitations on explainability, consistency, and expansibility, which may only achieve suboptimal performances. In this paper, we propose a novel **E**xplainable generative multi-modality **A**lignment method for transferable **Rec**ommender systems, i.e., **EARec**. Specifically, we design a two-stage pipeline to achieve unified multi-modality alignment of items and the sequential recommendation task, respectively. In the first phase, we present a generation task that parallel aligns each modality from multiple source domains to an anchor with explainable meaning. Three modality features share the same anchor to achieve a consistent alignment direction. Additionally, we incorporate behavior-related information as an independent modality into the alignment framework, establishing a bridge that promotes the alignment between multi-modalities and behavior. In the second stage, we composite the aligned modality encoders into a unified one and then transfer it to the target domain to enhance sequential recommendation. The pipeline that adopts parallel multi-modal alignment and composition shows flexibility and scalability for incorporating new modalities. Experimental results on multiple public datasets demonstrate the superiority of EARec over multi-modality recommendation baselines and further analysis indicates the explainability of generative alignment.

## Keywords

Transferable recommendation, Multi-modality alignment, Explainable alignment

ACM Reference Format:
Anonymous Author(s). 2024. Explainable Generative Multi-Modality Alignment for Transferable Recommendation. In . ACM, New York, NY, USA, 9 pages. https://doi.org/10.1145/nnnnnnn.nnnnnnn

## 1 Introduction

Conventional sequential recommendation methods model item representations based on item IDs, which are non-shared across

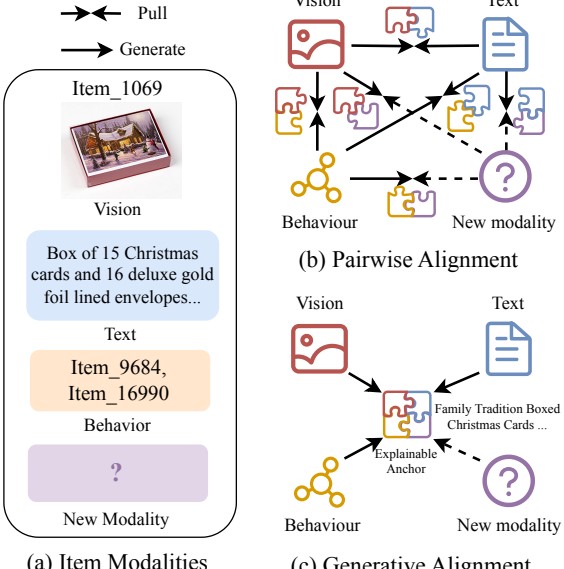

**Figure 1: Illustrations of (a) various modalities of an item; (b) pairwise alignment paradigm; (c) our proposed explainable generative alignment paradigm.**

domains and limit these models' transferability. In recent years, due to its cross-domain generalizability, multi-modality information has been used in item representation learning to achieve transferable recommendation [3, 8, 11, 19].[1]

Early transferable recommendation methods typically introduce a single modality to learn cross-domain universal item transition patterns [3, 8]. Subsequent studies [11, 19] investigate harnessing more modality types, achieving sufficient performance improvement. However, as shown in Figure 1(a), different modalities usually have distinct information richness, offering varied perspectives (e.g., color or style in vision, and brand or quantity in text) on an item. How to bridge the gap between modalities to promote universal multi-modality item representation learning become a new issue.

Some recent works attempt to utilize the widespread *pairwise* alignment paradigm to achieve the alignment of modalities [11, 12, 19, 27]. As illustrated in Figure 1(b), the pairwise alignment paradigm mitigates the gap between modalities by modeling the representation consistency between two modalities. Despite notable success, this alignment paradigm has limitations in the following three aspects: *Explainability* of the alignment process. **Firstly**, cross-modal pairwise alignment is typically performed based on the latent high-dimensional representations of two modalities. The

---

[1] Our work is related to 'User modeling, personalization and recommendation" track

aligned results are still abstract and difficult to understand. *Consistency* of the aligning direction. **Secondly**, the pairwise alignment of multiple modalities may lead to the direction inconsistency problem. For example, in Figure 1(b), the visual modality needs to be aligned simultaneously with the textual and behavioral modalities. Inconsistent alignment directions can lead to an unstable alignment process, thereby resulting in unreliable modality representations. *Expansibility* of new modalities. **Thirdly**, as shown by the dotted line in Figure 1(b), the new modality needs to be realigned with all existing modalities, which greatly increases the complexity of training and the difficulty of adapting to new modalities.

To alleviate the above limitation of pair-wise alignment, in this paper, we propose a **E**xplainable multi-modality **A**lignment method for transferable **Rec**ommender systems, **EARec**. Specifically, we achieve EARec by solving the following three challenges:

**Challenge 1**: How to design the alignment framework to ensure explainability, consistency, and expansibility? We propose an explainable generative alignment method, which aligns the multimodal information of items into a unified explainable space, as shown in Figure 1(c). Considering that large language models (LLMs) can comprehend different modality inputs and generate responses, we implement the generative alignment method based on LLMs. Specifically, we fine-tune the LLM to take inputs from different item modalities and generate the same output. This shared alignment objective across modalities is referred to as "anchor", which can be any unique item content, such as title or image. The consistency of this generative target facilitates the effective alignment of different modalities in subsequent model composition. Additionally, this method shows high explainability by allowing for the evaluation of alignment quality through the generated results.

**Challenge 2**: How to incorporate task-specific recommendation information into the alignment process? The ultimate goal of aligning modalities is to better represent items for recommendation tasks. To incorporate recommendation-specific signals into the alignment process, we treat recommendation behavior as a modality and integrate it into the alignment framework. Additionally, we add item relation information as an auxiliary signal in the generative alignment task.

**Challenge 3**: How to effectively utilize multiple modalities for distinct recommendation scenarios? In various recommendation scenarios, user preferences for different modalities can vary. For instance, in e-commerce platforms, users may focus more on the color and style of items (visual modality). In information stream recommendations, users may prefer items related to those they have just viewed (behavioral modality). Therefore, we adopt an adjustable modality composition method that adaptively adjusts the weights of different modalities for different recommendation scenarios, balancing the contributions of various modalities to downstream recommendation tasks and ensuring optimal performance.

To evaluate the effectiveness of our proposed EARec, we first collect item modality data from multiple domains and construct instruction samples to fine-tune the LLM through the generative alignment task. We then composite multiple LLMs that have undergone modality alignment. Subsequently, we transfer the model capable of simultaneously understanding multiple modalities to new recommendation domains. The model is used to obtain multimodal representations of items and these modality representations are fed into the recommendation model. Experimental results indicate that, aided by the aligned item modality representations, the performance of downstream recommendation tasks achieved significant improvements.

The main contributions of our work are summarized as follows:

- We investigate a novel multi-modality alignment paradigm to alleviate the limitations of the existing pairwise alignment approach in recommendation scenarios.
- We propose an explainable multi-modality alignment method that aligns multiple modalities into a unified explainable space through shared generative alignment objectives. We incorporate recommendation-related information during the alignment process to achieve the aligned multi-modal representations conducive to recommendation tasks.
- Experimental results demonstrate that leveraging the multi-modal representation generated by explainable multi-modality alignment can effectively enhance recommendation performance.

## 2 Related Work

**Transferable Recommendation** Transferable recommendation is a popular research area within recommendation systems, aiming to explore the effective transfer of knowledge learned from the source domain to the target domain to alleviate issues of data scarcity or cold start in the target domain. Early works often assumed an overlap of users or items between the source and target domains, using this overlap as a bridge to connect the two domains [9, 18, 22, 28]. Recently, some research [3, 7, 8, 11, 19] has begun to explore unified item representations to enhance the transferability of recommendation systems by using modality information. In particular, items are represented solely through modalities without relying on non-generalizable ID information across domains. Based on this, models can be constructed to leverage large amounts of modality data from multiple domains to learn universal item representation patterns. Then, the trained model is transferred to new domains to improve recommendation performance. However, these methods have not deeply investigated the gaps that exist between modalities, limiting the full utilization of modality information. Although MISSRec [19] and PMMRec [11] consider the issue of aligning modalities, they only adopt traditional pairwise alignment paradigms, which suffer from vague alignment direction, poor explainability, and difficulties in introducing new modalities.

**Multi-modal Recommendation.** Multi-modal information is prevalent in the interactions between recommendation systems and users, playing a crucial role in user decision-making. In recent years, various works have explored the incorporation of multi-modal information into user preference modeling. Early methods introduce modality information as auxiliary features or constructing modality-specific graphs for feature aggregation [6, 20, 23]. Some recent approaches attempt to address the issue of modality gaps and utilize self-supervised learning to achieve cross-modal alignment [14, 27]. However, these methods typically introduce modality features based on ID embeddings, thereby limiting them to single recommendation domains and lacking transferability.

**Multi-modal Learning and Alignment** Multi-modal learning has rapidly developed in fields such as computer vision (CV) and

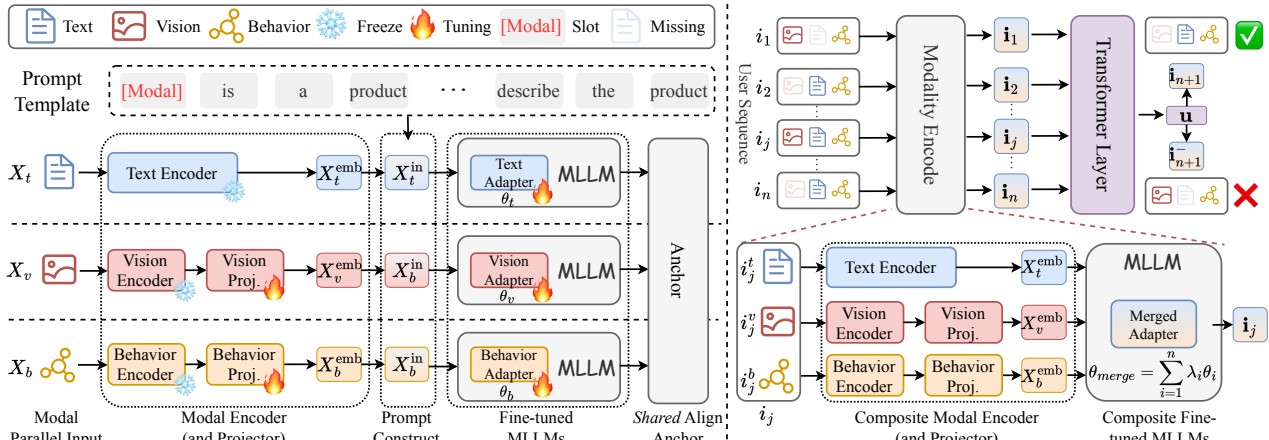

**Figure 2: The overall framework of our proposed EARec, which consists of two stages for source and target domain, respectively. In stage 1, we fine-tune multiple MLLM to align three modalities, i.e., Text, Vision, and Behavior, to a shared meaningful anchor in a parallel way. In stage 2, the fine-tuned MLLMs are composited first and then serve as a modality encoder to generate the multi-modality item representations to facilitate sequential recommendation.**

natural language processing (NLP), particularly with the rise of multi-modal large language models (LLMs), which have brought transformative changes to multi-modal learning paradigms. Common multi-modal large models enable LLMs to process both text modalities and a new modality simultaneously. This is typically achieved by first aligning modalities with text and then using modality data for instruction tuning [1, 13]. Another research direction explores enabling a single multi-modal large language model to handle multiple modalities beyond text. This can be realized by utilizing modality encoders that have inherent alignment across various modalities [4] or by fine-tuning the LLM with instruction data containing multiple modality inputs [26]. Recently, model composition approaches have made significant progress in alignment effectiveness and scalability [2, 16]. However, the aforementioned methods primarily focus on general domains, and detailed research on multi-modal learning in the context of recommendation systems—especially regarding how to handle the unique behavioral information in recommendations—remains insufficient.

## 3 Method

### 3.1 Framework Overview

The overall framework of our proposed EARec is illustrated in Figure 2. Considering $n$ different modalities $\{m_1, m_2, \ldots, m_n\}$, our goal is to align them into a unified explainable representation space, thereby obtaining a model capable of simultaneously understanding multiple modalities. Specifically, we design a two-stage pipeline to achieve unified multi-modality alignment of items and the sequential recommendation task, respectively.

In the first stage, we develop a generative alignment method to align the inputs of different modalities into a unified explainable space. Specifically, we input various modalities into the customized multi-modal large language model (MLLM) and fine-tune it using

the same generative objective. The strategy of parallel alignment ensures that the MLLM can fully understand and model each modality. Notably, we regard the item behavior information as an independent modality to integrate the recommendation-specific signal into MLLM. In the second stage, we draw inspiration from model composition methods [2] and composite multiple fine-tuned MLLMs to obtain a unified MLLM that can simultaneously understand different modalities. In particular, we merge the parameters of the MLLMs and integrate modal-specific components (i.e., modal encoders and projectors) into a unified framework. The composited MLLM is then utilized for the recommendation task. Due to its ability to simultaneously understand different item modalities, the composited MLLM can accept multiple modal inputs of items and generate a unified item representation. These representations can further be employed to derive user sequence representations, facilitating the prediction of the next item. Following previous works [19], Our stage 1 is conducted on multiple mixed source domain data, followed by the application of the trained MLLM in Stage 2 for recommendations in the target domain.

Next, we will introduce the explainable modality alignment method in Section 3.2 and the recommendation method based on aligned MLLM in Section 3.3.

### 3.2 Explainable Modality Alignment

*3.2.1 Unified Generative Alignment.* To achieve alignment between different modalities and mitigate the discrepancies that exist among them, we propose an explainable generative alignment method. Unlike the commonly used pairwise alignment methods based on contrastive learning, our approach employs explainable alignment objectives. We independently conduct the alignment processes for different modalities, training $n$ models $\{M_1, M_2, \ldots, M_n\}$ that can understand various modalities. The advantage of this alignment approach is that it avoids alignment conflicts and exhibits good

A chat between a curious human and an artificial intelligence assistant. The assistant gives helpful, detailed, and polite answers to the human's questions.

**Instruction**

USER: <image> is a product in the Amazon ecommerce platform, the category of this product is Office Products. <image>, <image> are often bought together with this product. Please describe the product. ASSISTANT:

**Prompt**

Family Tradition Boxed Christmas Cards - Set of 15. Greeting Cards. Vermont Christmas Company.

**Response**

**Figure 3: The instruction template of explainable generative alignment task. The response part is the shared generation objective for each modality.**

scalability for new modalities. To ensure that the subsequent composition effectively integrates the understanding capabilities of different models for different modalities, we employ the same alignment objective, referred to as an anchor point, during the alignment training process of the $n$ models, thereby ensuring that these models comprehend the modalities within the same Explainable space.

Specifically, recognizing the potential of LLMs to understand different modalities and generate feedback, we fine-tune the LLM to learn to comprehend various modalities of items. For a given modality $m$, we construct instruction samples as input for the LLM, as illustrated in Figure 3, where the response portion represents the shared alignment objective throughout the alignment training process for all modalities. For the constructed prompt $X_m^{in}$, we denote the modality-specific slot as $X_m$ and the modality-independent prompt portion as $X_p$. For the different modality inputs, we employ corresponding modal encoders for vectorization, for instance, visual modalities can be encoded using models like CLIP [17]. For modality representations that is not same with the dimensions of the LLM, we map them through the corresponding modal projectors. Formally, we express this as:

$$X_m^{emb} = [MoProj(MoEnc(X^m))]$$

$$X_m^{in} = [X_m^{emb}, MoEnc(X^p)]$$

where $MoEnc$ is the modal encoder and $MoProj$ is the mapping function for modality representations. Note that for text modalities, $MoEnc$ essentially represents the word embedding of the LLM and does not require a projector. Next, we input $X_m^{in}$ into the LLM and train it using an autoregressive generative task, defined as:

$$\max_\theta \sum_{(x,y)\in\mathcal{Z}} \sum_{t=1}^{|y|} \log\left(P_\theta\left(y_t \mid x, y_{<t}\right)\right)$$

where $x$ comprises the instruction and prompt portions in $X_m^{in}$, $y$ represents the response portion, $y_t$ is the $t$-th token of $y$, $y_{<t}$ includes all tokens preceding the $t$-th token, $Z$ denotes all instruction data, and $\theta$ represents the parameters of the LLM. Following previous work, we only train a subset of weights within the LLM, which we refer to as the adapter.

Through this generative task, we align the heterogeneous modality inputs into a unified feature space comprehensible to the LLM, providing a highly explainable alignment process. We can evaluate the alignment effectiveness by comparing the feedback generated by the LLM with the actual feedback, a capability that traditional pairwise alignment methods struggle to achieve.

*3.2.2 Recommendation-aware Alignment.* To enhance the utility of aligned modal representations for downstream recommendation tasks, we incorporate recommendation information into the alignment process from two perspectives. On one hand, we treat item behavior as a distinct item modality, reflecting the behavioral relationships among items. Understanding this modality can significantly aid in recommendation tasks. Specifically, we utilize the item embedding from an existing recommendation system as the modal encoder for the item behavior modality, a practice that holds practical significance and can be well integrated with widely used recommendation systems. On the other hand, we introduce relationships among items into the instructions of the alignment task. As illustrated in Figure 3, in addition to the images of the items themselves, two other images of items that share a co-purchase relationship are also included as part of the prompt to help the model better comprehend the items. This relationship among items is typically introduced in prior work through knowledge graphs, employing graph embedding techniques to optimize item representations. Our approach provides a more flexible means of incorporating this knowledge to enhance the LLM's understanding and representation of modalities.

## 3.3 Recommendation with Aligned Modality

*3.3.1 Model Composition.* Through the generative alignment training in Section 3.2, we obtain $n$ LLMs $\{M_1, M_2, \ldots, M_n\}$ that understand different modal information. Since these modalities share the same alignment anchor points during the alignment process, these MLLMs share a common Explainable space for different modalities, allowing us to composite these models to integrate their understanding capabilities across various modalities.

Specifically, in the model composition, two components need to be addressed. One part consists of modality-related components, such as the modal encoders and projectors, which serve to vectorize modal information and map it into the LLM's Explainable space. The other part comprises modality-independent components, namely the parameters of the LLM. In the model composition, we retain the encoders and projectors for different modalities, enabling us to handle inputs containing multiple types of modal information. For the $n$ LLMs, we merge the parameters of their respective modal adapters.

Considering that the importance of different modalities varies across different downstream recommendation scenarios, it is essential to adaptively adjust the attention given to different modalities in the item modal representation for various contexts. To achieve this, we adopt an adaptive weight model composition method. Specifically, when merging the parameters of $n$ MLLMs, we adjust the parameter weights corresponding to different modalities, formally expressed as:

$$\theta_{merge} = \sum_{i=1}^{n} \lambda_i \theta_i$$

where $\lambda_i$ represents the adaptive weights. In practice, the selection of $\lambda_i$ can be determined by evaluating the performance of the merged model on downstream datasets in alignment tasks.

*3.3.2 Downstream Recommendation.* By combining multiple MLLMs, we obtain an MLLM capable of simultaneously understanding different item modalities, which we refer to as EARec. With the introduction of behavioral modalities and item relationship knowledge, EARec can effectively encode various modalities of items and serve the downstream recommendation process. We evaluate the transfer recommendation performance of EARec in a sequence recommendation task based solely on item modality representations. Specifically, for a user's time-ordered interaction sequence $s = \{i_1, i_2, \ldots, i_n\}$, the multimodal representation of item $i_j$ is obtained through the following formula:

$$\mathbf{i}_j = \text{EARec}([i_j^t; i_j^v; i_j^b])$$

where $\text{EARec}(\cdot)$ denotes the hidden state at the last position of the model's final layer as the multimodal representation of the item. Here, $i_j^t, i_j^v, i_j^b$ represent the behavioral, textual, and visual modalities of the item, respectively. Notably, our method is not limited to these three modalities, as the proposed alignment framework can effectively extend to new modalities; it only requires completing the generative alignment task and then incorporating the model. Additionally, for any item $i_j$, there may be cases where a certain modality is absent; however, since EARec does not require simultaneous input of modality data during alignment, it can effectively address the issue of modality absence.

Subsequently, we follow prior work by employing a transformer layer to aggregate the item representations from user interactions to obtain sequence representations. Specifically, the input to the model is the sum of the multimodal representation of the item $\mathbf{i}_j \in \mathbb{R}^d$ and the absolute positional embedding $\mathbf{p}_j \in \mathbb{R}^d$:

$$\mathbf{f}_j^0 = \mathbf{i}_j + \mathbf{p}_j.$$

The entire sequence $\mathbf{F}^0 = [\mathbf{f}_1^0; \ldots; \mathbf{f}_n^0] \in \mathbb{R}^{n \times d}$ is then input into $L$ layers of transformer layers, where the output of the $l + 1$ layer is:

$$\mathbf{F}^{l+1} = \text{FFN}(\text{MHAttn}(\mathbf{F}^l)).$$

We take the hidden state at the last position of the $L$ layer, $\mathbf{f}_n^L \in \mathbb{R}^d$, as the representation of the user sequence $\mathbf{u} \in \mathbb{R}^d$.

Finally, the prediction score for the next item is obtained by calculating the inner product between the user sequence representation and the candidate item representation:

$$\text{score}_{(i_{t+1}|s)} = \text{Softmax}(\mathbf{u} \cdot \mathbf{i}_{t+1}).$$

During training, we utilize cross-entropy loss to optimize the next item prediction task, and during evaluation, we rank the candidate items based on the inner product scores. It is noteworthy that all parameters of the EARec model remain frozen during the training process, allowing us to offline obtain multimodal representations for all items, thereby ensuring that the downstream recommendations achieve comparable efficiency to traditional recommendation methods.

**Table 1: Statistics of the datasets after preprocessing. "Avg. n" denotes the average length of item sequences.**

| Datasets | #Users | #Items | #Image | #Inters. | Avg. n |
|---|---|---|---|---|---|
| **Stage 1** | 1,361,408 | 446,975 | 94,151 | 14,029,229 | 13.51 |
| - Food | 115,349 | 39,670 | 29,990 | 1,027,413 | 8.91 |
| - CDs | 94,010 | 64,439 | 21,166 | 1,118,563 | 12.64 |
| - Kindle | 138,436 | 98,111 | 0 | 2,204,596 | 15.93 |
| - Movies | 281,700 | 59.203 | 8,675 | 3,226,731 | 11.45 |
| - Home | 731,913 | 185,552 | 34,320 | 6,451,926 | 8.82 |
| **Stage 2** | | | | | |
| - Office | 87,436 | 25,986 | 16,628 | 684,837 | 7.84 |
| - Arts | 45,486 | 21,019 | 9,437 | 395,150 | 8.69 |
| - Instruments | 24,962 | 9,964 | 6,289 | 208,926 | 8.37 |
| - Movielens | 610 | 3,650 | 1,846 | 89,664 | 147.99 |

## 4 Experiments

In this section, we first introduce the experimental setup, followed by presenting the experimental results and analyses.

### 4.1 Experiment Setting

*4.1.1 Datasets.* In Stage 1, we utilize the item modality information from five domains to perform the explainable generative alignment tasks on multiple MLLMs. Subsequently, in Stage 2, we apply the EARec composited by these MLLMs to derive multi-modal representations of items in downstream datasets, followed by performing the sequential recommendation. Specifically:

- **Stage 1 dataset**: We select five datasets from Amazon e-commerce dataset [5, 15] for the explainable generative alignment task in stage 1 of EARec, namely "Grocery and Gourmet Food", "Home and Kitchen", "CDs and Vinyl", "Kindle Store", and "Movies and TV".
- **Stage 2 dataset**: For downstream recommendations, we select three additional datasets from Amazon to evaluate EARec's transfer recommendation performance across domains, namely "Office Products", "Arts, Crafts and Sewing" and "Musical Instruments". To evaluate the transfer performance on a new platform, we select a cross-platform dataset, i.e., Movielens[2].

For all datasets, following prior work [8, 19], we remove users and items with fewer than five interactions and organize the items according to the temporal order of user interactions. We consider three item modalities: Text, Vision, and Behavioral. Notably, our method can conveniently extend to accommodate any new modalities. The statistics of the datasets are summarized in Table 1.

*4.1.2 Baselines.* EARec is compared with the following baselines:

- **SASRec** [10] employs a self-attention mechanism to aggregate item ID embeddings in user sequences without incorporating additional modality information.
- **SASRec_T** is an extension of SASRec, utilizing item textual modality information to obtain item representations instead of ID embeddings.

---

[2]https://grouplens.org/datasets/movielens/latest/

**Table 2: Downstream recommendation performance of different models. The best, second-best and third-best performances are denoted in bold, underlined, and wavy-line fonts, respectively. The subscript "T", "V", and "B" denote the Text, Vision, and Behavior modality used in the item encoding. The superscripts $^*$ and $^{**}$ indicate $p \leq 0.05$ and $p \leq 0.01$ for the paired t-test of the best EARec variant vs. the best baseline.**

| Setting | | Baselines | | | | | Ours | | |
|---|---|---|---|---|---|---|---|---|---|
| Dataset | Metric | SASRec | $\text{SASRec}_T$ | $\text{UniSRec}_T$ | $\text{MoRec}_T$ | $\text{MISSRec}_{TV}$ | $\text{EARec}_{TV}$ | $\text{EARec}_{TVB}$ | Improv. |
| Office | HR@10 | 0.1064 | 0.1043 | 0.1046 | 0.1096 | 0.1038 | 0.1210 | **0.1234**$^{**}$ | 12.59% |
| | HR@50 | 0.1641 | 0.1709 | 0.1751 | 0.1794 | 0.1701 | 0.1973 | **0.1981**$^{**}$ | 10.42% |
| | NDCG@10 | 0.0710 | 0.0640 | 0.0627 | 0.0673 | 0.0666 | 0.0707 | **0.0713** | 0.42% |
| | NDCG@50 | 0.0835 | 0.0785 | 0.0780 | 0.0825 | 0.0808 | 0.0873 | **0.0876**$^{**}$ | 4.91% |
| Arts | HR@10 | 0.1074 | 0.1078 | 0.1099 | 0.1101 | 0.1119 | 0.1224 | **0.1244**$^{**}$ | 11.17% |
| | HR@50 | 0.1986 | 0.2050 | 0.2118 | 0.2127 | 0.2100 | 0.2329 | **0.2330**$^{**}$ | 9.54% |
| | NDCG@10 | 0.0571 | 0.0613 | 0.0602 | 0.0637 | 0.0625 | 0.0664 | **0.0671**$^{**}$ | 5.34% |
| | NDCG50 | 0.0769 | 0.0825 | 0.0823 | 0.0860 | 0.0836 | 0.0905 | **0.0908**$^{**}$ | 5.58% |
| Instruments | HR@10 | 0.1126 | 0.1175 | 0.1087 | 0.1229 | 0.1201 | 0.1241 | **0.1252**$^{**}$ | 1.87% |
| | HR@50 | 0.2087 | 0.2224 | 0.2079 | 0.2278 | 0.2218 | 0.2336 | **0.2362**$^{**}$ | 3.69% |
| | NDCG@10 | 0.0618 | 0.0690 | 0.0622 | 0.0717 | **0.0771** | 0.0667 | 0.0727 | - |
| | NDCG@50 | 0.0826 | 0.0917 | 0.0837 | 0.0944 | **0.0988** | 0.0909 | 0.0967 | - |
| Movielens | HR@10 | 0.0967 | 0.0803 | 0.0721 | 0.0557 | 0.0885 | 0.0984 | **0.1033**$^{**}$ | 6.83% |
| | HR@50 | 0.2852 | 0.2705 | 0.2705 | 0.2246 | 0.2361 | **0.2984**$^{**}$ | 0.2852 | 4.63% |
| | NDCG@10 | 0.0419 | 0.0352 | 0.0308 | 0.0249 | 0.0393 | 0.0440 | **0.0466**$^{**}$ | 11.22% |
| | NDCG@50 | 0.0826 | 0.0761 | 0.0740 | 0.0617 | 0.0703 | **0.0868**$^{**}$ | 0.0858 | 5.08% |

- **UniSRec** [8] learns cross-domain universal sequence patterns through item textual modality representations and employs MoE to adaptively adjust item representations in different domains.
- **MoRec** [24] incorporates item text modality and performs end-to-end optimization on the modality encoder and recommendation model.
- **MISSRec** [19] is based on item textual and visual modality representations, employing a multi-modal interest-aware module and cross-attention mechanisms to learn multi-interest user sequence representations.

*4.1.3 Evaluation Metrics.* We utilize two widely used evaluation metrics, HR@K and NDCG@K, to assess the performance of the downstream recommendation tasks. $K$ is set to 10 and 50. Following prior work [8, 10], we adopt a leave-one-out method to split the dataset. Specifically, for a user's interaction sequence, we use the last item for testing, the second-to-last item for validation, and the remaining items for training. We obtain the ranking list through the dot product scores between the user sequence representation and all items, and we report the average results over all users.

*4.1.4 Implement Details.* In Stage 1, we implement the explainable multi-modality generative alignment based on the transformer library [21] and the DeepSpeed library [3]. The backbone model is Vicuna-v1.5 [4]. We select three modalities input for alignment: item description, item image, and item embedding of recommendation model, with the alignment anchor set as the item title. The

text encoder is the LLM's word embedding, the vision encoder is *clip-vit-large-patch14-336* [5], and the behavior encoder utilizes the pre-trained item embedding from SASRec. We load the parameters of LLaVA's LLM part and vision projector before training for the vision modality. For behavior modality, we perform a continuous alignment on in-domain behavior data to alleviate the gap between domains. During training, we employ LoRA to efficiently fine-tune MLLM, the hyperparameter $r$ is set to 128, and $\alpha$ is set to 256. The learning rate for the LoRA parameters is set at $2 \times 10^{-4}$, while the learning rate for the projector is $2 \times 10^{-5}$. We conduct experiments on four NVIDIA RTX 3090 GPUs, with a global batch size of 16.

In Stage 2, we implement downstream recommendation tasks using the RecBole library [25]. During the model composition phase, the range of modality-adaptive adjustment weights is set to [0,1], ensuring that the sum of the weights equals 1. In the downstream recommendation phase, the number of layers and heads of the transformer encoder is set to 2. For all downstream recommendation experiments, we employ the Adam optimizer and carefully search for hyperparameters, with a batch size of 2048 and NDCG@10 as the evaluation metric, employing a patience of 10 for early stopping. We adjust the learning rate within the set {0.0003, 0.001, 0.003, 0.01} and the embedding dimension within {64, 128, 300}. The code is available at: https://anonymous.4open.science/r/EARec

## 4.2 Performance Comparison

We compare two variants of the proposed method, $\text{EARec}_{TV}$ and $\text{EARec}_{TVB}$, with several baseline methods. The difference between the $\text{EARec}_{TV}$ and $\text{EARec}_{TVB}$ is the input modality in item encoding.

---

[3] https://github.com/microsoft/DeepSpeed
[4] https://huggingface.co/lmsys/vicuna-7b-v1.5

[5] https://huggingface.co/openai/clip-vit-large-patch14-336

**Table 3: Analysis on modality expansibility in Office dataset. "w/o" denotes removing the alignment of specific modality. The best and second-best performances are denoted in bold and underlined fonts, respectively.**

| Model | HR@10 | HR@50 | NDCG@10 | NDCG@50 |
|---|---|---|---|---|
| EARec | **0.1234** | **0.1981** | **0.0713** | **0.0876** |
| - w/o Behavior | 0.1212 | 0.1972 | 0.0704 | 0.0870 |
| - w/o Vision | 0.1204 | 0.1951 | 0.0691 | 0.0853 |
| - w/o Text | 0.1200 | 0.1940 | 0.0692 | 0.0854 |
| - w/o All | 0.1189 | 0.1940 | 0.0699 | 0.0862 |

The overall experimental results are shown in Table 2. From the results, several observations can be made.

First, recommendation methods incorporating modal information generally outperform traditional methods, i.e., SASRec. This demonstrates that introducing modal information effectively enhances item representation and improves recommendation tasks. Second, although introducing more modal information and using a pairwise alignment approach to reduce gaps between modalities, MISSRec still underperforms MoRec, which only uses a single modality. It indicates that the pair-wise alignment does not integrate the modalities effectively and impairs the recommendation performance. Third, the benefits of transferable recommendation baseline models are not pronounced in cross-platform datasets, i.e., Movielens, suggesting the limitation of only performing transferable learning from the same platform.

The two variants of our method, $EARec_{TV}$ and $EARec_{TVB}$, achieve the best overall performance. Specifically, $EARec_{TV}$ outperforms MISSRec in most cases. This indicates that, compared to traditional pairwise alignment methods, our proposed explainable generative alignment method is more effective in incorporating modalities. Furthermore, $EARec_{TVB}$ achieves better performance than $EARec_{TV}$ by utilizing the behavior modality for item representation, demonstrating the effectiveness of our method in expanding to new modalities and showcasing the potential to enhance recommendation performance through further incorporating modalities. The performance improvements on the Movielens dataset indicate that our method has learned more generalizable modality representations, leading to better transfer recommendation performance.

### 4.3 Analysis of Modality Expansibility

In this section, we analyze the modality expansibility of EARec. We evaluate the impact of incorporating a new modality and the overall effect of aligning multiple modalities on the final performance. Specifically, we compare EARec with four of its variants: (1) **w/o Behavior**, (2) **w/o Vision**, (3) **w/o Text**, and (4) **w/o All**, i.e., the original model before the integration of all three modalities. To ensure a capacity for modality handling, we directly adopt LLaVA for **w/o All** variants.

The comparison results are shown in Table 3, from which several observations can be drawn. First, the incorporation of each new modality improves the model's performance, with the Vision and Text modalities contributing relatively more to the improvements.

Wing Sung 3008 Transparent Piston Fountain Pen Set of 4 Pieces , 4 Color Pen Cap. Fountain Pens. erofa.

**Item Title**

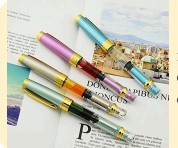

is a product in the Amazon ecommerce platform, the category of this product is Office Products. Please describe the product.

**Image Prompt**

**LLaVA**: The product in the image is a set of five different colored pens, which are part of the Amazon ecommerce platform's Office Products category. The pens are of the brand Pupius. **(RougeL: 0.26)**
**EARec**: 5 Pcs/lot Fountain Pen Pens with Ink Cartridges and Pens Cases. Fountain Pens. Zhuhai. **(RougeL: 0.41)**

**Generative Results**

Wing Sung 3008 Transparent Fountain Pen. Different from the traditional sucking way, this fountain pen uses rotational movement of the piston ink adding method, which is more convenient and efficient. Starting innovative, convenient and efficient life style. is a product in the Amazon ecommerce platform, the category of this product is Office Products. Please describe the product.

**Text Prompt**

**LLaVA**: Wing Sung 3008 Transparent Fountain Pen. Office Products. Wing Sung. **(RougeL: 0.51)**
**EARec**: Wing Sung 3008 Transparent Fountain Pen. Fountain Pens. Wing Sung. **(RougeL: 0.65)**

**Generative Results**

<Item_21935> is a product in the Amazon ecommerce platform, the category of this product is Office Products. Please describe the product.

**Behavior Prompt**

**LLaVA**: nobody is perfect, and we all make mistakes... **(0, *failed response*)**
**EARec**: 1 X 100ml Bottle of Black Ink for Fountain Pen. Fountain Pens. Zhenzhen. **(RougeL: 0.24)**

**Generative Results**

**Figure 4: The generative results of the EARec and LLaVA on various modality inputs. The key aspect of specific modality captured by EARec is highlighted with red font.**

Second, the original model, without the proposed explainable generative alignment across the three modalities, performs significantly worse, demonstrating the effectiveness of the proposed method.

### 4.4 Analysis of Explainable Alignment

In this section, we analyze the explainability of the proposed multi-modality alignment method. EARec aligns information from different modalities into a unified explainable space through generative alignment, allowing the alignment results to be compared with the anchor points to demonstrate the effectiveness of the alignment. Since we use item titles as anchor points, we evaluate the quality of the alignment by computing the RougeL score between the generated alignment results and the anchors. Additionally, we further analyze the correlation between the alignment results and downstream recommendation performance.

We illustrate the generative result of the alignment from modalities to anchor in Figure 4. By comparing the generative results of EARec and LLaVA, we find that EARec generates the response closer to the anchor text, as reflected by its higher RougeL score. More importantly, EARec captures item-specific characteristics unique to different modalities. In the visual modality, EARec provides text related to "Ink Cartridges" and "Pen Cases", demonstrating a deeper understanding of this modality. In the text modality, EARec extracts finer-grained item categories from the description, i.e., "Fountain

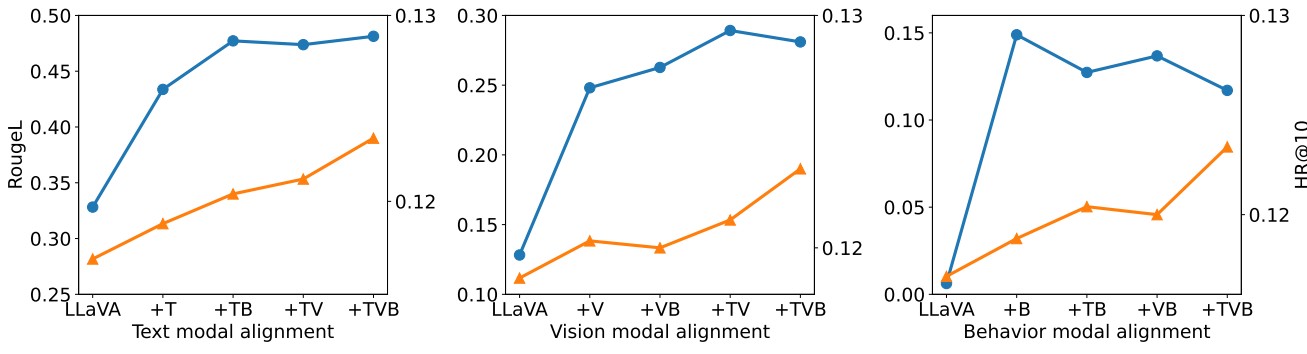

**Figure 5: RougeL of generative alignment vs. Downstream recommendation task in Office dataset. "+T","+V", and "+B" denote performing an alignment on Text, Vision, and Behavior modalities, respectively.**

Pens", instead of the broader "Office Products" given by LLaVA. Most notably, for outputs of the behavior modality, EARec generates text of a complementary item, i.e., "Bottle of Black Ink". This indicates that EARec can capture the item relation knowledge in item embedding, which is highly beneficial for recommendation.

Furthermore, we present the correlation between explainability and recommendation performance. As shown in Figure 5, we present the alignment performance for the three modalities and the corresponding downstream recommendation results. Several insights can be drawn from the figure. First, applying generative alignment to the vanilla model, i.e., LLaVA, significantly improves the RougeL scores of the generated results (e.g., from 0.3282 to 0.4337 for the Text modality), demonstrating the effectiveness of the alignment task. Notably, before aligning the Behavior modality, LLaVA was entirely unable to understand this modality, resulting in a near-zero RougeL score. Second, when the models for the three modalities are composited (i.e., LLaVA+TVB), the composite model shows further improvements in RougeL scores for the text and vision modalities, reflecting the model composition effectively integrates the model's ability to understand the three modalities. Nevertheless, this trend does not appear in behavior modality. We speculate the reason is the overfitting of the LLaVA+B variant since the training data of behavior modality is relatively insufficient compared to the other two modalities. Third, the alignment results' RougeL scores are generally proportional to the recommendation performance. This demonstrates that evaluating the alignment results can help us select the most suitable model for downstream recommendation tasks, highlighting the explainability of our model.

## 4.5 Analysis of Modality Adaptable Adjustment

In this section, we analyze the impact of the proposed modality-adaptive adjustment method on model composition. We conducted experiments using the Office dataset, adjusting the weights of the parameters associated with the three modalities in the MLLM to modify the model's understanding of different modalities, and we compared the corresponding downstream recommendation performance. The experimental results are shown in Table 4.

From these results, we observe that as the weight of the behavior modality parameters increases, the model's performance steadily

**Table 4: Modality Adaptable Adjustment weights for model composition in Office dataset. The best and second-best performances are denoted in bold and underlined, respectively.**

| Text | Vision | Behavior | HR@10 | HR@50 | NDCG@10 | NDCG@50 |
|------|--------|----------|-------|-------|---------|---------|
| 33% | 33% | 33% | 0.1187 | 0.1900 | 0.0674 | 0.0830 |
| 20% | 40% | 40% | 0.1185 | 0.1908 | 0.0703 | 0.0861 |
| 15% | 42.5% | 42.5% | 0.1192 | 0.1919 | 0.0703 | 0.0861 |
| 10% | 45% | 45% | 0.1193 | 0.1921 | 0.0722 | 0.0880 |
| 5% | 47.5% | 47.5% | 0.1194 | 0.1919 | 0.0721 | 0.0879 |
| 5% | 5% | 90% | **0.1234** | **0.1981** | **0.0713** | **0.0876** |

improves. This enhancement can be attributed to two primary reasons. First, since the understanding of the behavior modality is more complex than that of the text and vision modalities, increasing the weights of the parameters related to the behavior modality emphasizes the model's capability to comprehend it. Second, the behavior modality is specific to the recommendation task and contains more information beneficial for recommendations, making the prominence of this modality effective in improving the model's recommendation performance.

## 5 Conclusion

In this paper, we propose EARec, a novel explainable generative multi-modality alignment method for transferable recommender systems. Addressing the limitations of conventional pairwise alignment strategies, EARec leverages a two-stage pipeline to unify the alignment of diverse modalities and enhance sequential recommendation. It aligns multiple modalities to a shared anchor with explainable meaning, ensuring consistent alignment across modalities and incorporating behavior-related information as an independent modality. In the second stage, we composite aligned modality encoders to enable effective transfer to the target domain for improved recommendation performance. Experimental results on multiple datasets demonstrate the effectiveness of EARec, and further analysis shows its high explainability and expansibility.

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

Received 20 February 2007; revised 12 March 2009; accepted 5 June 2009

