# OpenReview forum: "Explainable Multi-Modality Alignment for Transferable Recommendation"
_ACM.org/TheWebConf/2025/Conference — WWW 2025 Poster_

### Official Review · Reviewer_4ew1 · 2024-11-18

**Novelty:** 6
**Technical Quality:** 6

**Review:**

The EARec method proposed in this paper is highly innovative.
1. By designing a two-stage pipeline and adopting an interpretable generative alignment method, multimodal information is aligned to a unified interpretable space, which solves the limitations of traditional pairwise alignment methods in terms of interpretability, consistency, and scalability. In particular, considering recommendation behavior as a modality and integrating it into the alignment framework, as well as adopting an adaptive weight model combination method, is a novel attempt in the field of multimodal recommendation.
2. Using the ability of a large language model (LLM) to process different modal inputs and generate a unified output as an alignment anchor provides a new idea for multimodal alignment, which is different from existing methods such as contrastive learning.

**Questions:**

My comments are mainly as follows:
1. Although the article mentions the use of technologies such as LoRA to improve training efficiency, the overall model is still relatively complex. Using multiple large language models (MLLM) for fine-tuning and combination may face high computing costs and resource requirements in practical applications, especially in large-scale data sets and high-concurrency recommendation scenarios. The model involves the processing and alignment of multiple modalities, such as text, images, and behaviors. Each modality requires a corresponding encoder and processing steps, which increases the complexity and computational overhead of the model.

2. The experiments are mainly focused on the Amazon and Movielens datasets. Although these datasets are representative, the data types and user behavior patterns are relatively limited. For other types of recommendation scenarios (such as social recommendation, news recommendation, etc.), the performance and applicability of EARec have not been fully verified.

**Reviewer Confidence:**

3: The reviewer is confident but not certain that the evaluation is correct

**Scope:**

4: The work is relevant to the Web and to the track, and is of broad interest to the community

---

### Official Review · Reviewer_bgTJ · 2024-12-01

**Novelty:** 4
**Technical Quality:** 3

**Review:**

The paper proposes a novel explainable generative multi-modality alignment method (EARec), effectively addressing the limitations of traditional pairwise alignment methods in terms of directional consistency, scalability, and explainability. The authors construct a two-stage framework consisting of a modality alignment phase and a recommendation transfer phase, integrating behavioral modality as a recommendation-specific signal, resulting in a modular and extensible design. The introduction of anchors for generative alignment adds theoretical significance and practical value. Through generative alignment tasks and RougeL score evaluation, the paper demonstrates the explainability of EARec, with experimental results showing that the generated outputs capture the relationships between modalities. Experiments on multiple datasets indicate that EARec significantly outperforms baseline methods in metrics such as HR@K and NDCG@K, particularly showcasing strong transferability in cross-platform recommendation scenarios.

Before considering publication, several issues must be addressed. If the following points are satisfactorily resolved, this reviewer believes that the fundamental contributions of this work are important to transferable multi-modal recommendation.

1. Expand the literature review to include existing studies on generative alignment methods in other domains, highlighting the originality of the proposed method in the context of recommendation systems.
2. The paper lacks a detailed explanation of the defined training losses, such as the loss functions used in pretraining tasks and cross-entropy loss during training. The calculation of the loss L should be explicitly introduced using equations.
3. While the model achieves notable recommendation performance, it lacks strong theoretical underpinnings. There is limited discussion on the theoretical basis behind generative alignment, and the paper lacks a detailed academic analysis of the method’s core effectiveness.
4. In Table 2, clearly indicate which models utilize which modalities (e.g., ID, V, and T), similar to the tables used in MISSRec, to improve clarity.
5. The comparison against baselines is insufficient. Add additional baselines to further validate the superiority of the model’s performance, such as other multi-modal models that incorporate behavior-related information in pretraining.
6. The performance of this method on the Instruments dataset is relatively poor. Please explicitly explain the reasons for its suboptimal performance in this scenario.
7. In Figure 5, the different lines representing specific metrics are not labeled. Add a legend or other means to make the visualization clearer.
8. Improve the layout and aesthetics of the figures, ensuring that different components (e.g., V and T, LLaVA and EARec) are visually more distinct.
9. Another major issue with the paper is the lack of sufficient explanation for the experimental results. You need to provide a detailed analysis of your simulation results and why you obtained such results. For example, in Section 4.2, explain why adding the behavioral modality improves recommendation performance.

**Questions:**

**Question 1**: Why is the loss function of the model not described in the form of computing L?
**Question 2**: Apart from the understanding by the LLM, how does the model specifically encode the behavioral modality of the items?
**Question 3**: How are the two lines in Figure 5 differentiated in terms of their meanings?
**Question 4**: Are there any baselines that incorporate behavioral information into pretraining? If so, why are they not marked explicitly? If not, does this introduce a certain degree of unfairness in the comparison? As far as I know, some models have already integrated user interaction information into pretraining.
**Question 5**: Regarding the suggestions I provided, do you have any questions or objections?

**Reviewer Confidence:**

3: The reviewer is confident but not certain that the evaluation is correct

**Scope:**

4: The work is relevant to the Web and to the track, and is of broad interest to the community

---

### Official Review · Reviewer_iTfr · 2024-12-01

**Novelty:** 3
**Technical Quality:** 4

**Review:**

To address the challenges of lacking explainability, consistency and expansibility in multimodal alignment, this paper presents an explainable generative multimodal alignment method for transferable recommendations, namely EARec. It first maps each modality, along with behaviour data, to an anchor with explainable meaning. Then, it compiles the aligned modality encoders into a unified one for transferable recommendations. Experimental results show the effectiveness of the proposed method.

This paper is overall well-organised and easy to follow. However, several limitations exist:
1. The motivation is not convincing enough, especially the part pairwise alignment "may lead to inconsistency problem". I suggest the authors provide more theoretical or empirical analysis to justify this claim.
2. In Figure 1 (c), the reason why the proposed method is "explainable" is still unclear.
3. Is this method specialised for transfer to sequential recommendations? If not, please include more empirical evaluations and analyses to discuss the effectiveness of the learned representations in other RS tasks.
4. Since visual modalities are encoded by models like CLIP, the visual and text representations are essentially mapped and aligned in a pairwise modality way by CLIP, which appears to contrast with the goal of this work.
5. Apart from visual and text modalities, how would other modalities, e.g., audio and video data, be encoded for the input of LLM?
6. The number of items with a co-purchase relationship is set to two in the methodology. Will a different setting lead to performance differences?

**Questions:**

1. The motivation is not convincing enough, especially the part pairwise alignment "may lead to inconsistency problem". I suggest the authors provide more theoretical or empirical analysis to justify this claim.
2. In Figure 1 (c), the reason why the proposed method is "explainable" is still unclear.
3. Is this method specialised for transfer to sequential recommendations? If not, please include more empirical evaluations and analyses to discuss the effectiveness of the learned representations in other RS tasks.
4. Since visual modalities are encoded by models like CLIP, the visual and text representations are essentially mapped and aligned in a pairwise modality way by CLIP, which appears to contrast with the goal of this work.
5. Apart from visual and text modalities, how would other modalities, e.g., audio and video data, be encoded for the input of LLM?
6. The number of items with a co-purchase relationship is set to two in the methodology. Will a different setting lead to performance differences?

**Reviewer Confidence:**

3: The reviewer is confident but not certain that the evaluation is correct

**Scope:**

3: The work is somewhat relevant to the Web and to the track, and is of narrow interest to a sub-community

---

### Official Review · Reviewer_tfFK · 2024-12-01

**Novelty:** 4
**Technical Quality:** 4

**Review:**

This paper proposes a generative multi-modality alignment framework for cross-domain recommender systems that can align modalities in a unified explainable representation space. The framework first fine-tunes n large language models (LLMs) on item modalities in source domain with a shared alignment objective. Then, the framework uses the n fine-tuned LLMs to generate representations for items in target domain and employs a transformer layer to generate the user sequence representations used for the downstream recommendation. The paper evaluates the proposed method on a small-scale public dataset and shows that it outperforms baseline methods in terms of HR and NDCG.


Pros:
* The paper is easy to follow and presented well.

Cons:
* The paper lacks novelty.
* The experiment is not adequate.

**Questions:**

1. The experimental setup is very similar to MISSRec, but the performance of MISSRec is significantly lower than reported in the original paper. Did you choose the best parameters for MISSRec to ensure a fair comparison?

2. The paper lacks novelty. As indicated in Tables 2-3, the improvements of EARec are mainly due to the LLMs. Incorporating modality alignment results in only a 0.003 improvement, while EARec w/o all shows a 0.01 improvement compared to other baselines that do not use LLMs. The improvement from your proposed modality alignment seems to be due to experimental variation.

3. The paper suggests that aligning modalities into a unified representation space with the same fine-tuning objective is possible, even with different LLMs. The authors are encouraged to include both theoretical and experimental evidence to support this claim.

4. The Amazon dataset used is small, which will include too much variation. Authors should use large-scale datasets to demonstrate effectiveness.

5. The experiment using MovieLens is flawed. Baseline methods do not use LLMs, making their generated representations less transferable. The transferability of the proposed method may only be due to LLMs. More details about the MovieLens data, such as the coverage of visual and text modalities, should be provided.

**Reviewer Confidence:**

4: The reviewer is certain that the evaluation is correct and very familiar with the relevant literature

**Scope:**

4: The work is relevant to the Web and to the track, and is of broad interest to the community